# Investigating the Pearl Millet (*Pennisetum glaucum*) as a Climate-Smart Drought-Tolerant Crop under Jordanian Arid Environments

Nabeel Bani Hani [1] , Fakher J. Aukour [2,*] and Mohammed I. Al-Qinna [2]

1   National Agricultural Research Center (NARC), Amman 19381, Jordan
2   Department of Land Management and Environment, Prince El Hassan Bin Talal Faculty of Natural Resources and Environment, The Hashemite University, Zarqa 13133, Jordan
*   Correspondence: fakagr67@hu.edu.jo

**Abstract:** To investigate drought tolerance under arid conditions, eleven pearl millet breeds (HHVBC tall B6; IP13150; IP19586; IP19612; IP22269; IP6110; IP7704; MC94C2; P. millet icms7709; Sudan-pop I; Sudan-pop III) were tested under arid water-scarce climatic conditions. A field randomized complete block design experiment with three replicates per year was conducted at the Deir-Alla Regional Agriculture Research center in the middle Ghor within the Jordan Valley from 2010 to 2020. The plant-deficit irrigation was maintained at 80% based on the crop water requirements using a time-domain reflectometer. The plant morphological characteristics, forage production, seed formation, and water-use efficiency (WUE) were monitored for ten years for two case scenarios: seed and forage production. The individual and combined drought indices of the precipitation, temperature, and vegetation were calculated and correlated with the millet morphological and yield parameters. Climate change analyses show significant impacts, reaching a 1 mm/year reduction in precipitation and a 0.04 mm/year increase in air temperature, which causes the study area to be more prone to drought events. Along with the proven increase in the drought intensity over time, the millet breeds showed significant drought-tolerance capacities under arid, drought-prone conditions by adjusting their system to tolerate salt, heat, and water stresses. For the seed production scenario, the WUE ranged from 27 to 57.3 kg/ha·mm, and from 7.1 to 14.9 kg/ha·mm for fresh and dry conditions, respectively. The IP13150 millet breed showed the highest capacity to tolerate the drought of Jordan's environment, and it is thus recommended as a good substitute under water-scarcity situations, with an average production of 17.7 ton/ha. For the vegetative production scenario, the WUE ranged from 32.03 to 64.82 kg/ha·mm for the fresh biomass and from 10.8 to 24.6 kg/ha·mm for the dry biomass. Based on the WUEs and vegetative production results, the IP19586, IP22269, IP19612, IP7704, and HHVBC tall B6 millet breeds are recommended as forage support due to their phenological characteristics, which tolerate drought and heat conditions. In contrast to the vegetation drought index, both the precipitation and temperature drought indices show strong correlations (above r > 0.6) with the plant growth factors and a moderate correlation (0.3 < r < 0.6) with the yield factors. Both precipitation and temperature indices are capable of explaining the variations among millet breeds, especially as related to millets' morpho-physiological characteristics.

**Keywords:** pearl millet; deficit irrigation; drought tolerance; fodder yield; water-use efficiency; adaptation

## 1. Introduction

Along with rapid population growth, the increasing energy-food-water demand, and sudden refugee migrations, drought remains the most complex natural phenomenon threatening the sustainable development of all sectors, including the economy, agriculture, water, ecosystems, health, tourism, and even society, at the global, regional, and local levels [1,2]. Over the last two decades, many researchers have pointed to the increase in

the frequency and severity of drought [3–5] that has adversely affected billions of people around the world.

Jordan is a developing country that is located in the MENA region with limited and fragile natural resources and is ranked as among the most water-deficit countries in terms of water availability [6–10]. The water-crisis threat is accelerating with the current and projected climate change impacts and extreme events, such as drought. The country's water scarcity and food insecurity continue to worsen because of high population growth and refugee flux, and it is accelerated by climate change. Moreover, it places additional pressure on the already existing scarce and depleted natural resources [11–14]. Because of competition among water-consuming sectors, the available irrigation water, especially in the Jordan Valley, has declined. Explicitly, the water-release portion for irrigation declined from 155 million cubic meters in 2003 to 130 million cubic meters in 2020 [15]. Despite the water-resource constraints, agricultural productivity has increased through the adoption of new farming and irrigation technologies and the use of climate-tolerant crops [16].

The sixth assessment report of the Intergovernmental Panel on Climate Change highlighted the likely increase in extreme events of droughts and floods across many regions [17]. Similarly, the future dynamic climate projections within the Third National Communication Report of Jordan to the UNFCCC predict that intense future droughts are extremely likely to occur [18]. The report also indicated that it is extremely likely that the average air temperature will increase from 2.1 to 3.6 C by the end of the century, while the precipitation is expected to reduce significantly by 15 to 20%.

Jordan has witnessed many extreme drought events, especially between 1958 and 1962, and through the period 1997–2000, and lately, in 2020 [19–23]. Jordan's government declared a state of drought in the country for the first time in 1999. The recorded Jordanian economic losses included 70% camel-herd loss in the period between 1958 and 1962, 30% of the sheep flocks by disease and malnutrition in the 1997 drought, a 40% reduction in red meat and milk, an 83% reduction in wheat production, a sharp drop in the dam water levels, and access was cut to regional surface water because of the severe drought in 1999 [18,24–26].

In response to the drought situation in Jordan, the government added drought as one of the disaster risks to be monitored and controlled by the Jordanian National Center for Security and Crisis Management (NCSCM), established in 2015. In 2017, Jordan's government delegated the responsibility for establishing and institutionalizing the national Disaster Risk Reduction (DRR) platform to the NCSCM. Additionally, an interministerial drought committee was recently established, holding six ministerial members and headed by the Ministry of Interior. However, the governmental authorities are still struggling with the emergency plans, and they are progressing slowly in developing the management/adaptation proactive and action plans to cope with the disastrous impacts of this hazard. Among these plans, the ministries are investigating the management options to be included in the drought strategies. Recently, Jordan's government established a Climate Change National Adaptation Plan; however, drought vulnerability and adaptation are ineffectively tackled. To ensure that proper drought preparedness and management plans are based on actual experimental findings, this research emphasized searching for drought-tolerant crops to mitigate the drought impacts and risks, taking millet as an example.

Pearl millet (*Pennisetum glaucum* (L.) R. Br.) is a C4 photosynthetic-pathway climate-resilient cereal crop. It is a cereal grain that is commonly known to belong to the grass family and the Poaceae family. It is an introduced, annual, warm-season crop and a summer annual forage crop consumed as food and as fodder for livestock feeding [27,28]. Pearl millet is a tall, warm season, annual grass that is drought and heat tolerant. It can grow well in soils with high salinity, low pH, and low fertility.

The Sahel region from Senegal to central Sudan is where millet originated, and it is now mostly grown in West Africa and Asia for both forage and grain production. Millets have long, scabrous leaves with a high leaf-to-stem ratio, and their solid stems are frequently adorned with dense hair [29–31]. Under ideal environmental conditions, the plant has

a tendency to tiller copiously and can compensate for uneven stand establishment. To support the developing plants, prop roots emerge from the lower nodes [32–34]. The Pearl Millet kind of millet has the biggest kernels overall. The color of the kernel can range from white to grey to pale yellow to slate blue to purple to brown. The plant has a four-meter height limit [35,36]. The amount of wheat, millet, canary seed, and other grains that Jordan imported from Argentina in 2018 was 15.13 thousand US dollars, according to the UN Comtrade Database on International Trade [37]. Due to the lack of improved variations, the introduction of millet into the plant species in marginal areas, and its usage as a rainfed crop, because it is a hardy plant, millet yields were seen to be lower in many nations.

Many studies are currently attempting to introduce new species with high productivity through the use of modified kinds that can multiply green fodder and, as a result, might feed twice as many animals per unit area as conventional fodder crops [38]. International research has been undertaken through a number of studies to examine the water-use efficiency of pearl millet in water-limited conditions [39,40]. According to certain research [30,41,42], a plant's rooting behavior impacts its capacity to use the soil's moisture reserves and how much salinity it can endure. Others have looked into the production of pearl millet in dry conditions and scenarios with changing climatic variables [43–47]. Pearl millet has been proposed as a Climate-Resilient Nutri-cereal for reducing covert hunger and ensuring nutritional security [44,48–52]. Assessing the impact of drought on pearl millet's morphological and yield responses in dry environments would help guide the selection of prospective drought-tolerant pearl millet cultivars for drought adaptation choices. This research is justified by the actual evaluation of the diversity and effectiveness of many pearl millet types utilizing lengthy historical drought indices. The primary goal of this study was to link several drought indicators with the growth and yield characteristics of various types of pearl millet. The key claim is that drought indicators can identify and track Pearl Millet's ability for drought tolerance by inferring temporal plant phonological alterations under heat and drought loads.

## 2. Materials and Methods

### 2.1. Site Description

An experiment was carried out at the Deir-Alla Regional Agriculture Research Centre in the Jordan Valley's middle Ghor, 50 km west of Amman. The Centre is located at 32°13′ E, 35°37′ N, and 224 feet below sea level (Figure 1). The climate in the area is semi-arid, with warm winters and hot summers and an average annual rainfall of 280 mm. The mean maximum annual temperature is 30 degrees Celsius centigrade, and the mean minimum annual temperature is 15 degrees Celsius centigrade, which sometimes goes to a few degrees in harsh winters. The mean maximum annual temperature is 30 degrees Celsius centigrade, and the mean minimum annual temperature is 15 degrees Celsius centigrade, which sometimes goes to a few degrees in harsh winters. The mean relative humidity ranges from 30% in summer to 70% in winter. This region is highly prone to drought with unevenly distributed rainfall. The long-term annual potential evaporation at the site is about 2200 mm.

The soil is located within the Ghor map unit representing a fine, mixed, hyper-thermic, deep family of Typic Ustochrepts. It is characterized by dark brown and dark yellowish brown (7.5 YR–10YR 4\4) deep (>80 cm) clay loam and light clay soils with weakly cracked surfaces and a strong medium sub-angular block structure. The soil is highly calcareous and non-saline, with slope gradients of <2%. More information on the soil is presented in Table 1. The available water resources at the site are the convey pressurized pipes from the King Talal Reservoir (KTR) at Zerqa River with moderate salinity ranging from 1.4 dS/m in winter to about 3.0 dS/m in summer, which may negatively impact the salt accumulation in soil profile and deterioration of land productivity. Due to climate change impacts, salinity is increasing over time, making potential deterioration more likely.

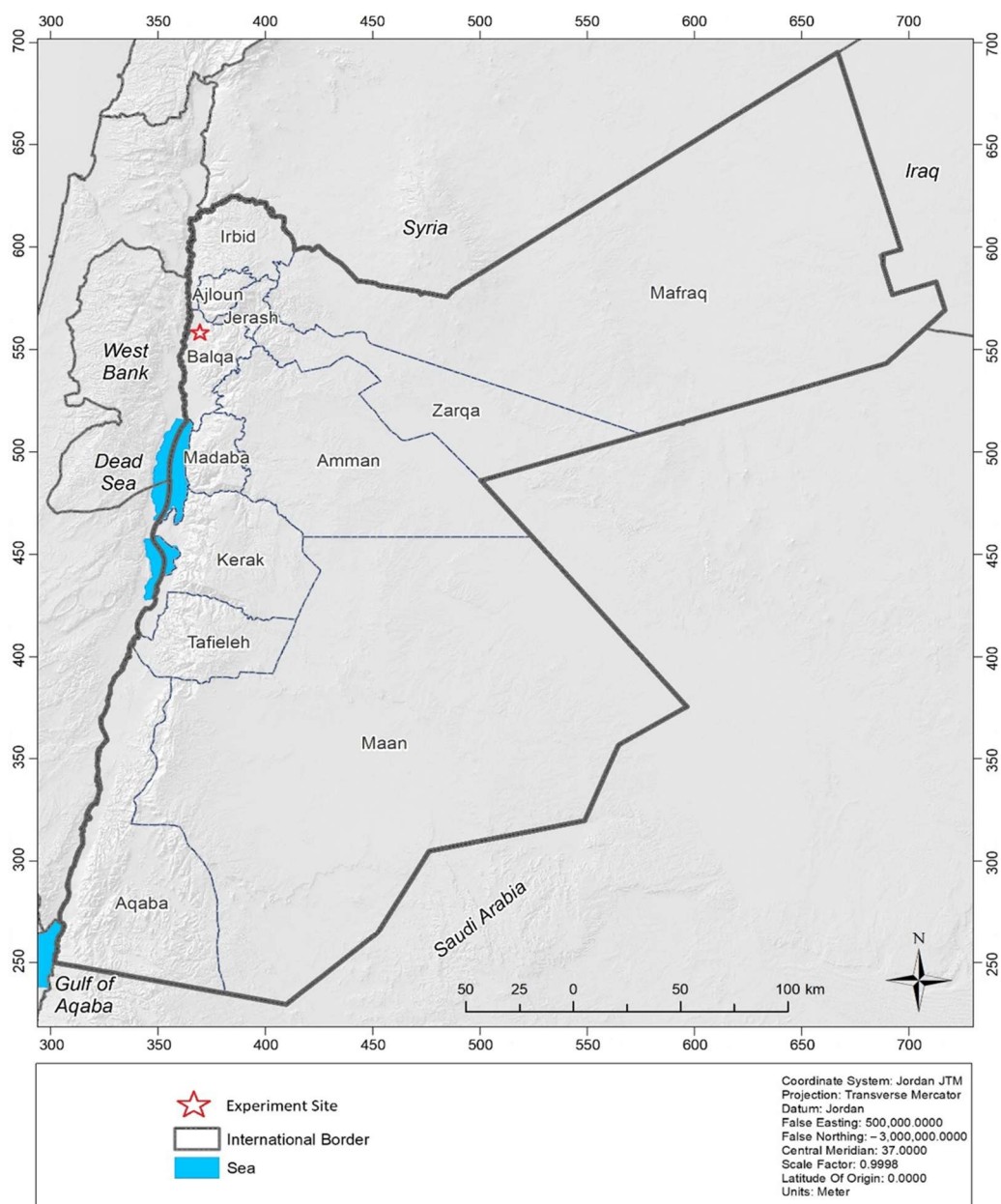

**Figure 1.** Experiment site of Deir-Alla Regional Agriculture Research center at the middle Ghor within the Jordan Valley.

**Table 1.** Soil Properties at Deir-Alla Regional Agriculture Research Center.

| Property | Unit | Depth (cm) 0–25 | Depth (cm) 25–50 |
|:---:|:---:|:---:|:---:|
| pH | | 7.8 | 7.9 |
| EC | dS/m | 3.43 | 1.6 |
| P | ppm | 35.6 | 25.9 |
| K | ppm | 573.9 | 504.6 |
| O.M. | % | 1.5 | 0.9 |
| $CaCO_3$ | % | 30.3 | 38.7 |
| N | % | 0.098 | 0.092 |

**Table 1.** *Cont.*

| Property | Unit | Depth (cm) 0–25 | Depth (cm) 25–50 |
|---|---|---|---|
| Clay | % | 40.3 | 41.4 |
| Silt | % | 43.3 | 40.8 |
| Sand | % | 16.4 | 17.8 |
| FC | % | 0.33 | 0.325 |
| PWP | % | 0.211 | 0.222 |
| WHC | % | 120 | 103 |
| Soil Texture | | Silty Clay | Silty Clay |

### 2.2. Drought Monitoring

Thirty years of daily climate data (1980–2020) were obtained from in-situ weather station. DrinC software version 1.5.73 was used to calculate the Drought Standardized Precipitation Index (*SPI*) [53]. The long-term monthly and seasonal rainfall records were initially normalized using the gamma distribution function. The temporal *SPI* magnitudes were calculated by dividing the difference between the normalized seasonal precipitation and its long-term seasonal precipitation mean by the standard deviation (Equation (1)).

$$SPI = (Xi - Xm)/\sigma \tag{1}$$

where *Xi* is the seasonal precipitation at the rain gauge station, *Xm* is the long-term seasonal mean (i.e., 30 years of records), and where $\sigma$ is its standard deviation.

According to McKee et al., a drought event occurs any time the *SPI* is continuously negative and reaches an intensity of −1.0 or less [54]. The event ended as *SPI* became positive. In this study, drought severity was divided into seven classes: Extremely wet (*SPI* > 2), very wet (1.5 to 1.99), moderately wet (1.0 to 1.49), near normal (−0.99 to 0.99), moderate drought (−1.49 to −1), severe drought (−1.99 to −1.5), and extreme drought (*SPI* < −2). In addition, the Combined Drought Index (*CDI*) was computed as a linear weighted average from the Precipitation Drought Index (*PDI*), Temperature Drought Index, and Vegetation Drought Index (*PDI*) (Equation (2), Equation (3), and Equation (4), respectively). Based on Balint et al. recommendations, the weights were assigned as 50% for *PDI*, 25% for each *TDI* and *VDI* [55].

$$PDI_{i,m} = \frac{\frac{1}{IP}\sum_{j=0}^{IP-1} P_{i,(m-j)}^*}{\frac{1}{(n\times IP)}\sum_{k=1}^{n}\left[\sum_{j=0}^{IP-1} P_{i,(m-j),k}^*\right]} \times \sqrt{\left(\frac{RL_{m,i}^{(P^*)}}{\frac{1}{n}\sum_{k=1}^{n} RL_{m,k}^{(P^*)}}\right)} \tag{2}$$

$$TDI_{i,m} = \frac{\frac{1}{IP}\sum_{j=0}^{IP-1} T_{i,(m-j)}^*}{\frac{1}{(n\times IP)}\sum_{k=1}^{n}\left[\sum_{j=0}^{IP-1} T_{i,(m-j),k}^*\right]} \times \sqrt{\left(\frac{RL_{m,i}^{(T^*)}}{\frac{1}{n}\sum_{k=1}^{n} RL_{m,k}^{(T^*)}}\right)} \tag{3}$$

$$VDI_{i,m} = \frac{\frac{1}{IP}\sum_{j=0}^{IP-1} NDVI_{i,(m-j)}^*}{\frac{1}{(n\times IP)}\sum_{k=1}^{n}\left[\sum_{j=0}^{IP-1} NDVI_{i,(m-j),k}^*\right]} \times \sqrt{\left(\frac{RL_{m,i}^{(NDVI^*)}}{\frac{1}{n}\sum_{k=1}^{n} RL_{m,k}^{(NDVI^*)}}\right)} \tag{4}$$

where $P^*$ is the modified annual precipitation, $T^*$ is the modified annual temperature, $NDVI^*$ is the modified annual average Normalized Difference Vegetation index, $IP$ is the interest period, $RRL_{m,i}^{(P^*)}$ (run-length) as the maximum number of successive years below long-term average rainfall in the intended period, $RL_{m,i}^{(T^*)}$ as the maximum number of successive years above the long-term average temperature, while $RL_{m,i}^{(NDVI^*)}$ as the

maximum number of successive years below long-term average *NDVI* in the *IP*, which indicates the number of years with relevant data, *j* is a summation running parameter covering the *IP*, and *k* is the summation parameter covering the years for which relevant data are available. The modified temperature, modified *NDVI*, and modified rainfall data were obtained using (Equations (5)–(8)) to avoid dividing by zero in certain cases, as rainfall in Jordan is mostly characterized by a distinct long dry season; this also helped to unify the ranges of the drought index values:

$$T^* = (T_{max} + 1) - T \tag{5}$$

$$RL^* = (RL_{max} + 1) - RL \tag{6}$$

$$NDVI^* = NDVI - (NDVI_{min} - 0.01) \tag{7}$$

$$P^* = (P + 1) \tag{8}$$

where *P*, *T*, and *NDVI* are the original precipitation, temperature, and *NDVI* values and *RL* is the original run-length.

The Normalized Difference Vegetation Index (*NDVI*) values were computed according to Kogan [56,57] using the ratio of responses in the near-infrared (*NIR*) and visible red portion of the spectrum (*R*) bands of the Advanced Very High-Resolution Radiometer (AVHRR) at the National Oceanic and Atmospheric Administration (NOAA) (Equation (9)).

$$NDVI = \frac{NIR - R}{NIR + R} \tag{9}$$

The extracted monthly *NDVI* data from the Terra Moderate Resolution Imaging Spectroradiometer (MODIS) Vegetation Indices (MOD13A3) Version 6 data at a 1 km spatial resolution [58]). A *CDI* of 1.0 thus represents average weather conditions; if the *CDI* is greater than 1.0, this represents wetter than average conditions, and if it is below 1.0, this represents dryer than average conditions. According to Balint et al. [55], five drought categories were implemented in this study, as presented in Table 2.

**Table 2.** Adopted *CDI* Drought Categories.

| Drought Category | *CDI* Value |
|---|---|
| No drought | >1.0 |
| Mild | 1.0–0.8 |
| Moderate | 0.8–0.6 |
| Severe | 0.6–0.4 |
| Extreme | <0.4 |

*2.3. Crop Experiment*

Eleven pearl millet breeds, including HHVBC tall B6, IP13150, IP19586, IP19612, IP22269, IP6110, IP7704, MC94C2, P. millet icms7709, Sudan pop I, and Sudan pop III, were cultivated at Deir-Alla Regional Agriculture Research Center from 2010 to 2020. Twelve hectares were used to set the field trials with a Complete Randomized Design (CRD) experiment, including three replications. The land seedbed preparation imitated the common practices adopted in the region and included chisel plowing followed by disk smothering for the top 20 cm. The seeds were planted in October and harvested at maturity by the end of February. The seeds were spaced 40 cm within rows and 40 cm between rows. Chemical fertilizers were applied prior to the plantation by 250 kg/ha triple superphosphate (46% $P_2O_5$) and 200 kg ha$^{-1}$ granular urea (46%N). At 30 to 40 cm plant height, another 200 kg ha$^{-1}$ of urea was applied through fertigation practice.

### 2.4. Crop Quality, Productivity, and Water-Use Efficiency

Two scenarios were used to evaluate the crop productivity and quality of millet breeds: Seed Production and Forage Production Scenarios. For the seed production scenario, the plants were solely tested under rainfed conditions. After maturity, the fruit length (cm), fruit diameter (cm), root depth (cm), mature fresh yield (ton/ha), mature dry yield (ton/ha), and seed yield (ton/ha) were measured per breed. A destructive 1.5 m × 1.5 m Quadrat sampling method was used to collect the plant samples after maturing. Random selection was used for plant aboveground sampling from each plot. The plant samples were chopped, fresh weighed, oven dried at 80 °C for 24 h, and dry weighed. Furthermore, the millet seeds were gathered by hand, fresh weighed, dried using a continuous flow drier for 48 h, and dry weighed. To investigate plants' reproductive efficiency, especially under heat and drought stresses, the Harvest Index (HI) was calculated as the ratio of grain yield to total aboveground biomass [59].

For the vegetative production scenario, all of the breeds underwent three cuts to investigate the progressive vegetative growth capacity (i.e., fresh forage at different cutting). This experiment was conducted under supplemental irrigations, in which the plants were irrigated using a drip system fitted by inline emitters of 4 l ph that are spaced 40 cm meters and 40 cm between lateral lines. The irrigation schedule was achieved once every ten days based on crop water demand.

The crop water requirements were estimated using the water balance equation (Equation (10)). Revised Penman-Monteith equation used to calculate the potential evapotranspiration (Equation (11)).

$$S = P + I - R - D - ET_o \tag{10}$$

$$ET_o = \frac{0.408 \times \Delta (Rn - G) + \gamma \left( \frac{900}{T + 273} \right) \times U_2 (e_s - e_a)}{\Delta + \gamma (1 + 0.34 \times U_2)}$$
$$\gamma = \frac{0.386 \times P}{L}, \ \Delta = 2 \times (0.00738 \times T + 0.8072)^7 - 0.00116 \tag{11}$$

where $S$ is the soil water stored in the root zone, $P$ is the precipitation, $I$ is the irrigation amount, $R$ is the runoff, $D$ is the drainage, and $ET_o$ is the potential evapotranspiration [mm/day], $Rn$ is the net radiation at the crop surface (MJ/m$^2$·day), $G$ is the soil heat flux density (MJ/m$^2$·day), $T$ is the mean daily air temperature at 2-m height (°C), $U_2$ is the wind speed at 2-m height (m/s), $e_s$ is the saturation vapor pressure (kPa), $e_a$ is the actual vapor pressure (kPa), $e_s - e_a$ is the saturation vapor pressure deficit (kPa), $\Delta$ is the slope vapor pressure curve (kPa/°C), and $\gamma$ is the psychrometric constant (kPa/°C).

The actual crop evapotranspiration ($ET_c$) is derived by multiplying the potential evapotranspiration ($ET_o$) by the crop factor ($K_c$) based on both growth stages and soil water measurements. Daily soil water content for the five soil increments (0–20, 20–40, 40–80, 80–120, and 120–160 cm) was measured at each site using the time domain reflectometer "Trime-FM3 TDR" (Imko GmbH, Ettlingen, Germany). Supplemental irrigation events were achieved regarding soil water deficit and crop water requirements. When the soil reaches the critical stage, irrigation was attained until reaching 80% of the available water. Throughout the temporal study, the amount applied varies from plot to plot and thus from replicate to replicate. The irrigation depth is the sum of the effective rainfall and irrigation minus the drainage water.

Crop measurements for the vegetative production scenario included the number of branches, plant height (cm), plant diameter (cm), fresh weight (ton/ha), and dry weight (ton/ha) per cut. The water-use efficiencies (WUE) were estimated by dividing the biomass weights (fresh biomasses and dry biomasses) per amount of water received (m$^3$), including the effective rainfall and supplemental water applied.

*2.5. Investigating the Impact of Drought on Plant Growth*

A Pearson Correlation Matrix was generated between the drought indices and millet production to investigate the drought indices' capacities and to represent the temporal variations among millet breeds (Equation (12)).

$$r_{xy} = \frac{\sum(x_i - \overline{x})(y_i - \overline{y})}{\sqrt{\sum(x_i - \overline{x})^2(y_i - \overline{y})^2}} \tag{12}$$

where $r_{xy}$ is the correlation coefficient, $x_i$ is the drought indicator at each specific year, $\overline{x}$ is the mean of the drought indicator across all the years of the study, $y_i$ is the millet production (ton/ha) for each year, and $\overline{y}_x$ is the millet production mean across all the years of the study.

A comparison between means for all crop parameters and among millet breeds was achieved using Tukey–Kramer test within JMP statistical software version 11.0.0 [60]. The Tukey–Kramer test represents an honest test at a 95% confidence level.

## 3. Results

*3.1. Climate Temporal Variability*

Recorded climate variables during the experiment period show temporal dispersity as indicated by the associated high standard deviations and the coefficient of variation, especially for precipitation (Table 3). The annual precipitation varies significantly from 99.8 mm during the dry period to 500 mm during the wet season (Figure 2).

Precipitation decreases insignificantly over time at a rate of 1 mm per year (*p*-value of 0.4732), thus suggesting a future reduction of about 25% by the end of the 21st century. Similarly, the annual and inter-annual air temperature varies significantly, ranging from 1.8 to a maximum of 50.1 °C (Figure 3). The trend of temporal change indicates a significant increase (*p*-value > 0.0001) by a rate of 0.04 °C per year, thus increasing up to 3.2 °C by the end of the 21st century.

**Table 3.** Preliminary statistical analyses of the climate variables during the study period.

| | Minimum | Maximum | Mean | Standard Deviation | Coefficient of Variation | Skewness | Kurtosis |
|---|---|---|---|---|---|---|---|
| Seasonal Precipitation | 99.8 | 500.6 | 280.9 | 102.22 | 36.39 | 0.420 | −0.30 |
| Daily Maximum Temperature | 8 | 50.1 | 30.1 | 8.05 | 26.70 | −0.26 | −1.17 |
| Daily Minimum Temperature | 1.8 | 36.3 | 18.1 | 5.60 | 30.91 | −0.14 | −1.13 |

*3.2. Drought Intensity and Severity*

All of the drought indicators confirm that the study area is highly prone to drought events. As indicated in Table 4, the precipitation drought indicator is the highest dispersed variable compared to temperature and vegetation drought indicators. This illustrates how the study area is subject to significant climate changes, especially in terms of precipitation followed by temperature.

The *SPI* trend in Figure 4 shows a frequent alternation from extremely wet (*SPI* > 2) to severe drought (−1.99 to −1.5) with the exception of lack of extreme drought (*SPI* < −2). Based on the *SPI* results, 14.6% of the period between the years 1980 to 2020 is classified as drought (*SPI* < −1), while 70.7% of the records are classified as near normal, and 14.6% are classified as wet. Local drought events' occurrence frequency tends to reach once every seven years, which persists from one to two consecutive years.

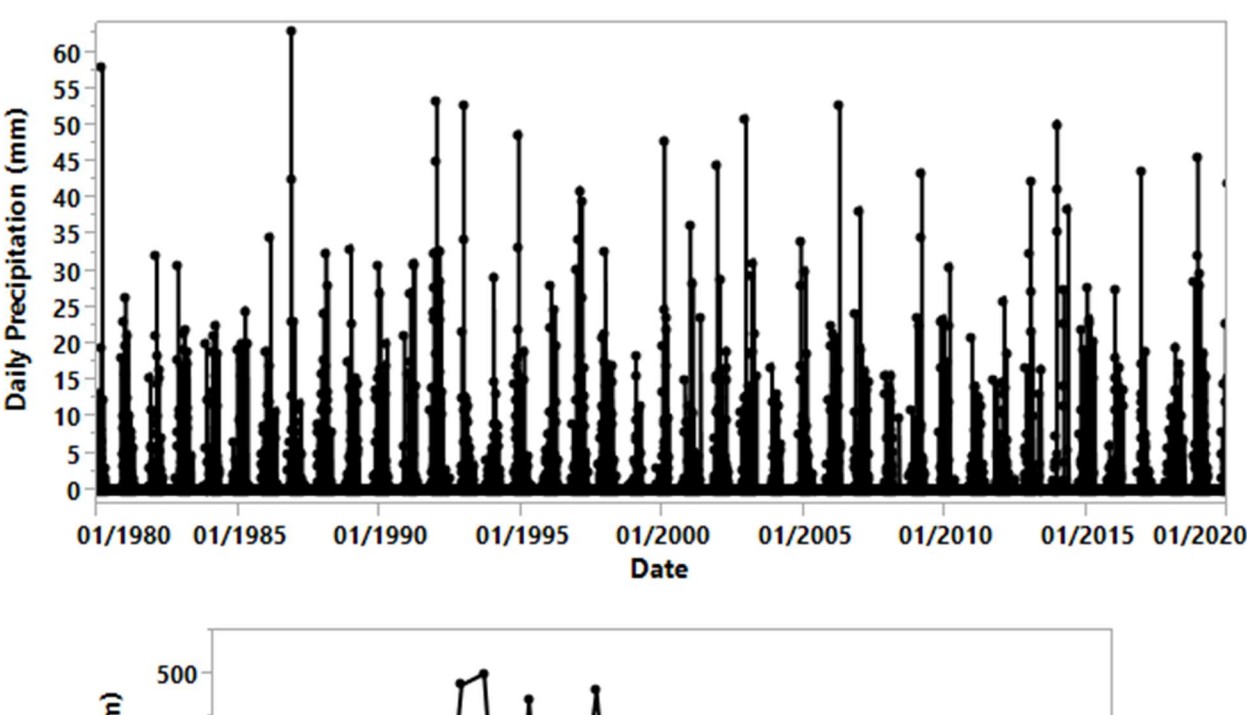

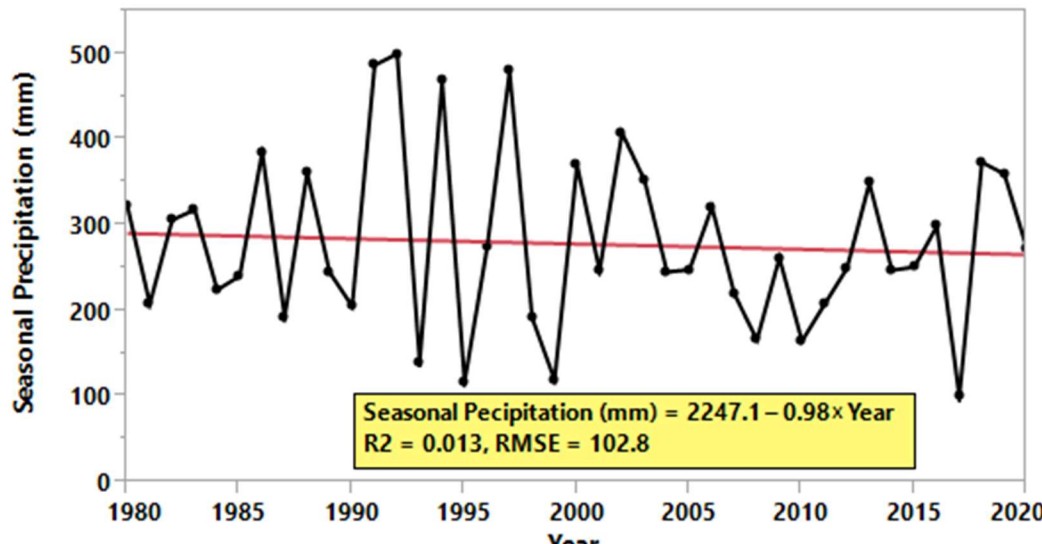

Seasonal Pecipitation (mm) = 2247.1 − 0.98 × Year
R2 = 0.013, RMSE = 102.8

**Figure 2.** Daily and Seasonal precipitation distribution and trend at Deir-Alla Regional Agriculture Research Center at the Middle Ghor within Jordan Valley.

**Table 4.** Drought Indices variability during the study period.

|  | Minimum | Maximum | Mean | Standard Deviation | Coefficient of Variation | Skewness | Kurtosis |
|---|---|---|---|---|---|---|---|
| Standardized Precipitation Index | −1.80 | 2.10 | 0.00 | 1.00 | 0.01 | 0.42 | −0.30 |
| Precipitation Drought Index | 0.16 | 2.15 | 0.99 | 0.46 | 46.28 | 0.58 | 0.33 |
| Temperature Drought Index | 0.30 | 1.90 | 0.97 | 0.31 | 31.80 | 0.15 | 1.14 |
| Vegetation Drought Index | 0.26 | 1.52 | 0.99 | 0.27 | 26.88 | −0.63 | 0.40 |
| Combined Drought Index | 0.52 | 1.62 | 0.99 | 0.26 | 26.77 | 0.60 | 0.01 |

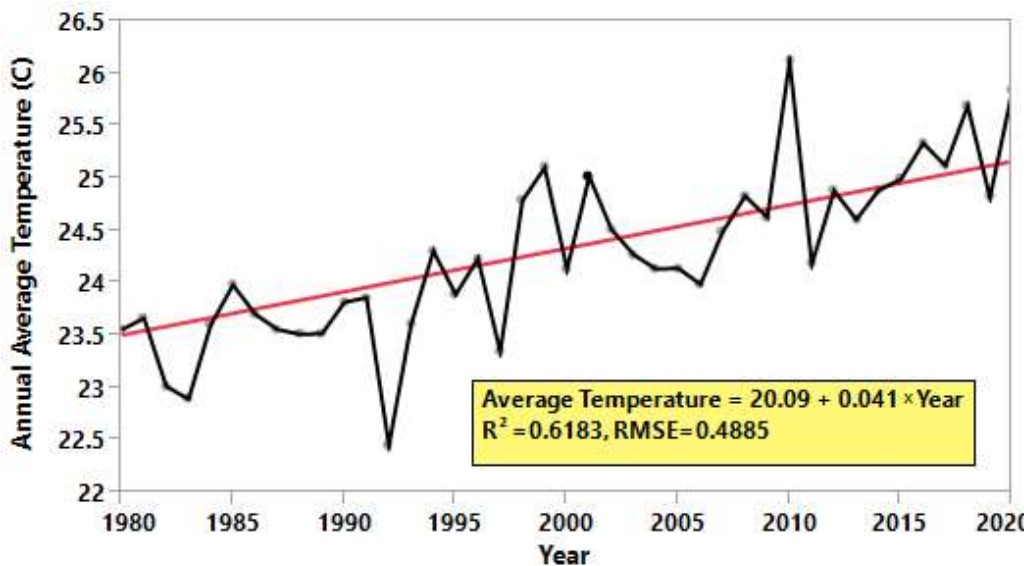

**Figure 3.** Daily temperature distribution and trend at Deir-Alla Regional Agriculture Research Center at the Middle Ghor within Jordan Valley.

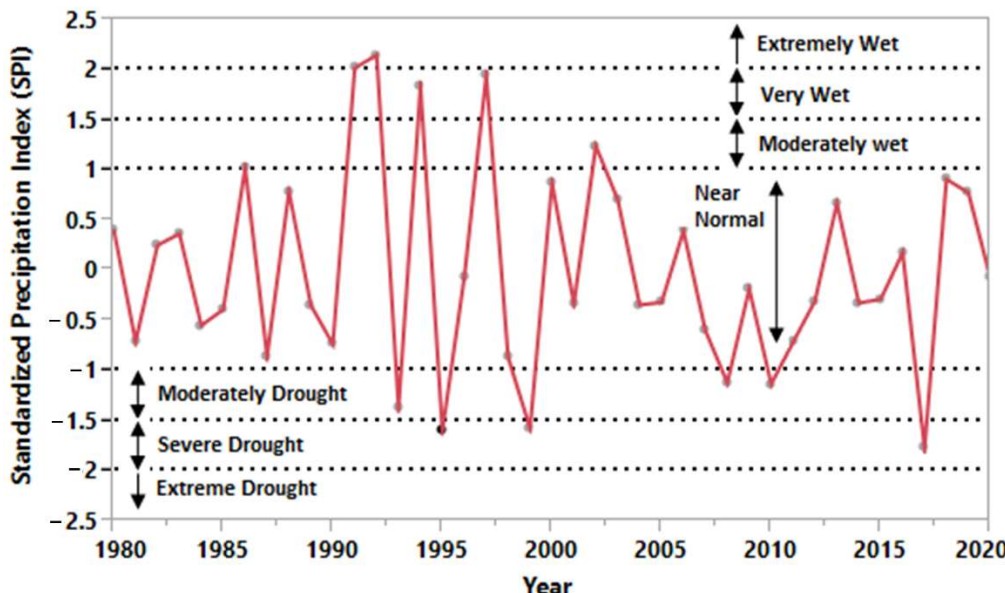

**Figure 4.** Temporal *SPI* distribution at Deir-Alla Regional Agriculture Research Center at the Middle Ghor within the Jordan Valley.

In terms of drought timing, the *PDI* always agrees with the *SPI* based on the temporal variation in the drought indicators (Figure 5). The highly severe precipitation droughts occurred mainly in 1995, 1999, and 2017, while severe precipitation droughts occurred in 1993, 2008, and 2010. On the other hand, the *TDI* temporal trend indicates that 2001 and 2010 were categorized as extreme temperature droughts, while 1999, 2007, and 2016 were categorized as severe temperature droughts. Finally, the *VDI* temporal trend shows that the year 2013 can be categorized as an extreme drought, while the years 1985 and 2017 can be categorized as severe drought events (Table 5). Although the *PDI*, *TDI*, and *VDI* show different magnitudes of temporal changes, they all have similar linear trends of increasing intensity and frequency over time. Figure 6 shows the increasing *CDI* rate is about 0.009 magnitude per year, thus shifting the drought mean by one level towards the severe category by 2050 and towards extreme droughts by 2100. This illustrates the severe threats of future climate impacts in the study area.

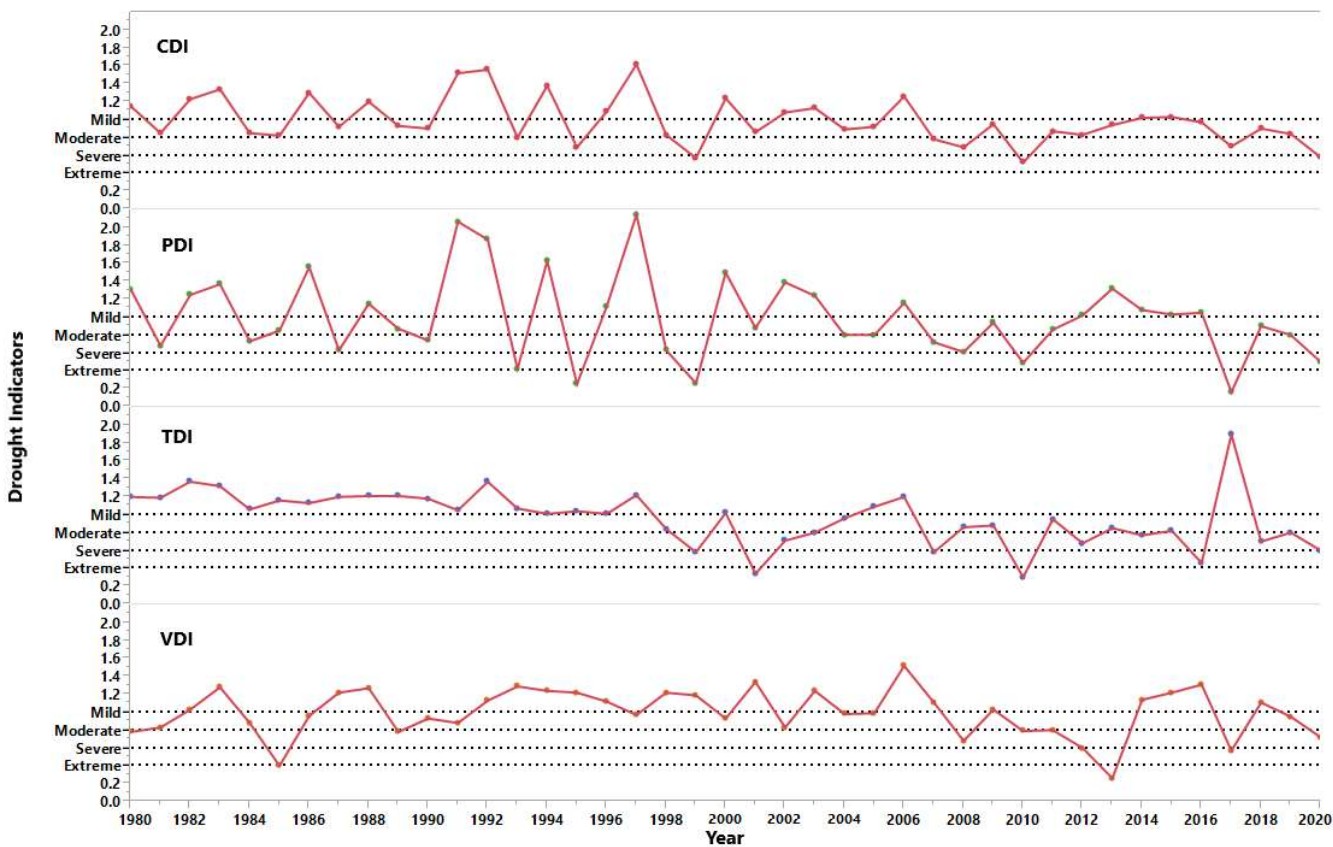

**Figure 5.** Temporal Drought Indicators distribution at Deir-Alla Regional Agriculture Research Centre at the Middle Ghor within the Jordan Valley.

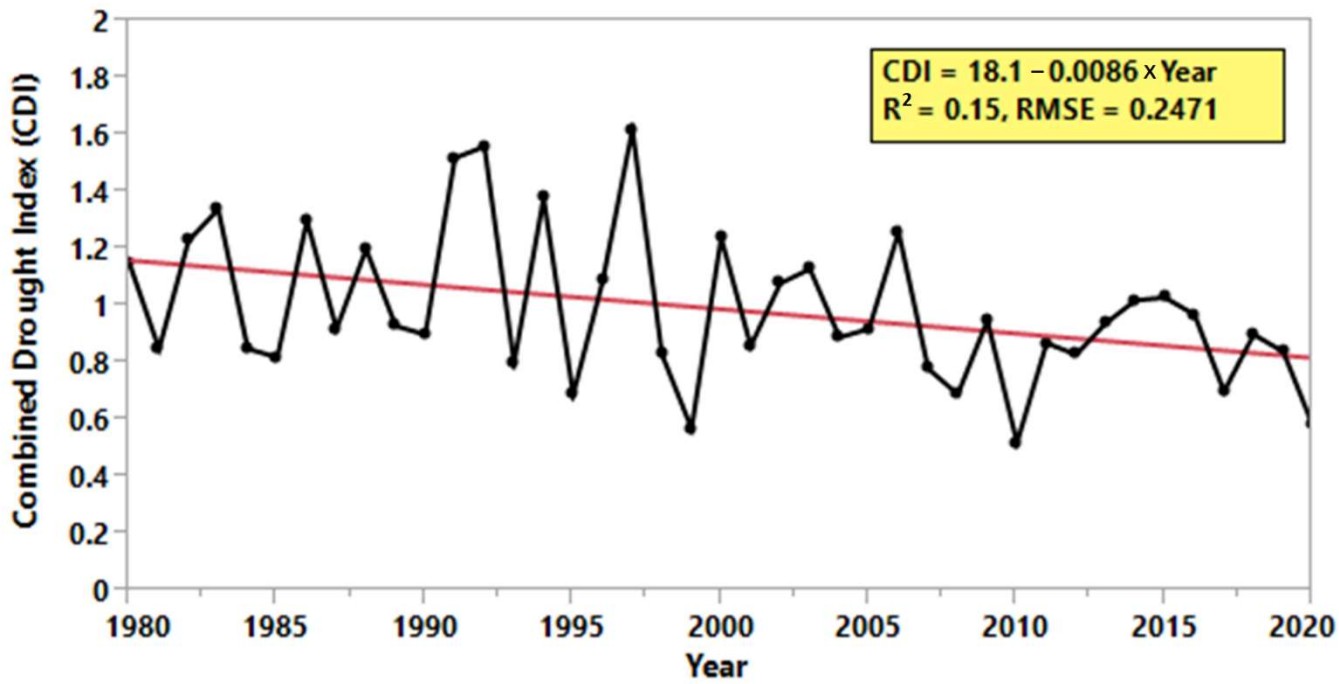

**Figure 6.** Temporal Trend of Combined Drought Indicator at Deir-Alla Regional Agriculture Research Centre at the Middle Ghor within the Jordan Valley.

**Table 5.** Drought Indices severity and categorization during the experiment period (2010–2020).

| Year | *PDI* | *TDI* | *VDI* | *CDI* |
|---|---|---|---|---|
| 2010 | 0.49 SD | 0.30 SD | 0.79 MD | 0.52 SD |
| 2011 | 0.86 MiD | 0.95 MiD | 0.80 MiD | 0.87 MiD |
| 2012 | 1.02 ND | 0.68 MD | 0.60 MD | 0.83 MiD |
| 2013 | 1.32 ND | 0.85 MiD | 0.26 ED | 0.94 MiD |
| 2014 | 1.08 ND | 0.77 MD | 1.14 ND | 1.02 ND |
| 2015 | 1.03 ND | 0.82 MiD | 1.22 ND | 1.03 ND |
| 2016 | 1.05 ND | 0.46 SD | 1.31 ND | 0.97 MiD |
| 2017 | 0.16 ED | 1.9 ND | 0.57 SD | 0.7 MD |
| 2018 | 0.90 MiD | 0.70 MD | 1.11 ND | 0.9 MiD |
| 2019 | 0.80 MiD | 0.8 MiD | 0.95 MiD | 0.84 MiD |
| 2020 | 0.50 SD | 0.60 MD | 0.72 MD | 0.58 SD |

ND is no drought, MiD is mild drought, MD is moderate drought, SD is severe drought, and ED is extreme drought.

### 3.3. Irrigation Water Variability

Based on the soil water budget determined using both TDR and crop needs, the amount of supplemental water applied is significantly different across the millet breeds (Figure 6) and across the study period (Figure 7), ranging from a minimum of 275 mm for a wet year to 365 mm in the driest year. On the other hand, some millet breeds display high water demands, as IP6110 and Sudan pop I regardless of the season (Table 6). Although the statistical analyses show no significant differences between water applied between breeds; however, the IP19586 and IP13150 millet breeds show low potential for water consumption and temporal variability as compared to other breeds.

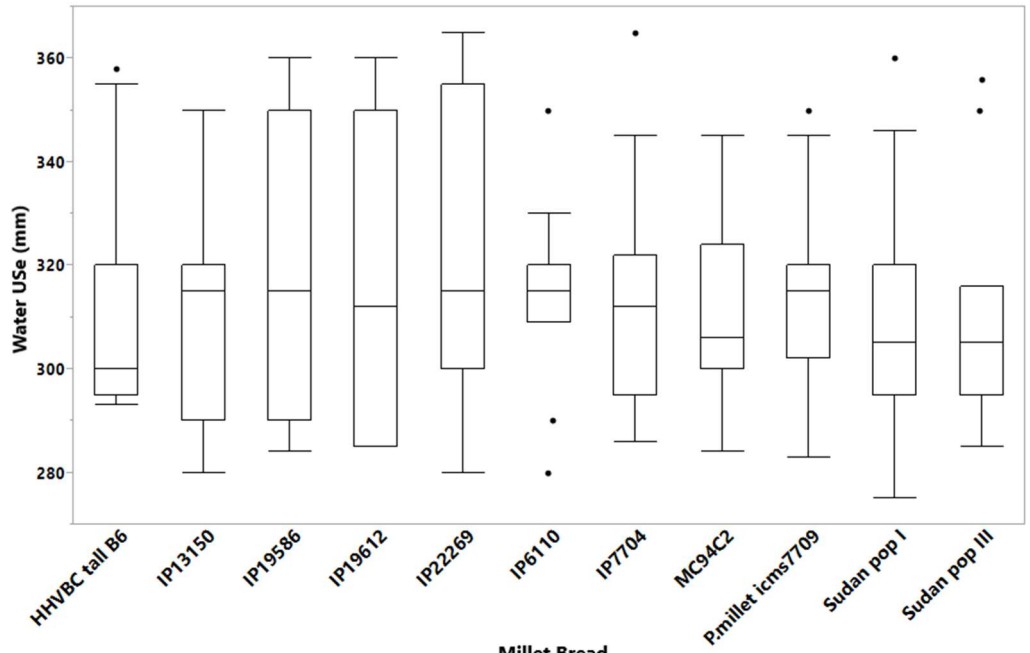

**Figure 7.** Variability in amount of water applied per millet breeds.

**Table 6.** Comparison between plant main parameters' means regarding Millet breeds.

| Millet Breed | Water Use (mm) | Number of Branches | Plant Diameter | Plant Height | Fruit Length | Fruit Diameter | Root Depth (cm) |
|---|---|---|---|---|---|---|---|
| IP22269 | 320 a | 13.55 a | 1.92 a | 147.09 b | 21.72 abc | 1.86 d | 120.2 b |
| IP19612 | 316 a | 7.00 f | 1.60 ab | 199.09 a | 19.34 bc | 2.16 bcd | 178.3 a |
| IP19586 | 315 a | 12.73 ab | 1.66 ab | 195.09 a | 18.37 bc | 2.03 cd | 175.2 a |
| IP6110 | 314 a | 9.73 cdef | 1.47 b | 204.45 a | 15.88 c | 2.51 abc | 180.1 a |
| IP7704 | 315 a | 11.82 abc | 1.56 ab | 210.91 a | 26.01 a | 1.79 d | 182.5 a |
| IP13150 | 311 a | 11.73 abc | 1.56 ab | 167.73 ab | 25.79 a | 2.79 a | 154.6 ab |
| HHVBC tall B6 | 313 a | 11.55 abcd | 1.55 ab | 196.73 a | 26.49 a | 2.74 a | 162.2 a |
| MC94C2 | 311 a | 8.91 def | 1.47 b | 200.64 a | 22.37 ab | 2.64 ab | 171.8 a |
| P.millet icms7709 | 316 a | 10.73 bcde | 1.47 b | 196.27 a | 18.30 bc | 1.86 d | 176.2 a |
| Sudan pop I | 311 a | 11.82 abc | 1.45 b | 195.55 a | 19.75 bc | 2.45 abc | 169.5 a |
| Sudan pop III | 313 a | 8.00 ef | 1.47 b | 195.73 a | 23.10 ab | 2.35 abcd | 162.6 a |
| Tukey–Kramer HSD | 32.741 | 2.7534 | 0.39817 | 47.529 | 5.8484 | 0.57131 | 55.344 |

Levels not connected by the same letter within the same column are significantly different at 95% confidence level.

### 3.4. Comparison of General Parameters among Millet Breeds

At a 95% confidence level, the comparison of the means for the number of branches, plant height, fruit length, fruit diameter, and root depth reveal significant differences. ($p < 0.001$) regarding millet breeds and seasons. The number of branches ranges from 16 for the IP22269 breed to only seven branches for the IP19612 millet breed (Table 6). Similarly, the recorded millet height ranges from 113 cm for the IP22269 breed to 271 cm for the IP7704 breed.

In terms of fruit length and diameter, the IP13150 and HHVBC tall B6 millet breeds have the best records, regardless of seasonal variations, with an average of 26 cm and 3 cm, respectively. Seasonal variations are caused by drought-related conditions, while breed variations are caused by phenology and physiology. Some pearl millet breeds, such as IP13150 and HHVBC tall B6, showed high records and low dispersion in terms of seasonal variations, indicating that they can withstand drought conditions. These breeds can adjust their phenology to rainfall patterns and are thus unaffected by drought conditions [61–63].

The root depth records show significant differences between breeds and seasons. The root depth varies from 120 cm for the IP22269 breed to 183 cm for the IP7704 breed. The inter-seasonal variation among pearl millet breeds, on the other hand, is very low, implying that pearl millet breeds can adjust their root system under drought conditions. This coincides with Ajithkumar et al., in which pearl millet breeds can adjust their root system to become longer under drought conditions [41]. Pearl millet produces deep and profuse root systems that can withstand drought effects by extracting water from the lower soil profile. On the other hand, Zegada-Lizarazu and Iijima indicated that higher WUE could explain the drought resistance of pearl millet [64].

### 3.5. Comparison of Millet Production and WUE among Millet Breeds

According to the first scenario, millet production varies greatly and significantly by breed and season. The average maturity fresh yield ranges from 8.3 to 17.7 ton/ha, with the maximum associated with the IP13150 millet breed (Table 7). Furthermore, the average dry yield ranges from 2.2 to 4.6 ton/ha, with the maximum associated with the IP13150 millet breed. The WUE ranges from 27 to 57.3 kg/ha·mm and from 7.1 to 14.9 kg/ha·mm for fresh and dry conditions, respectively. These values coincide with many researchers' findings for millet production under drip irrigation practices [14,40,65,66]. Despite temporal variations in millet production, as shown in Figure 8, the promising millet breed IP13150 could potentially be a good substitute in water-stress situations. ICBA reached similar conclusions, reporting that pearl millet salt-tolerant hybrids produced 11% more grain and 38% more fodder than the most widely cultivated dual-purpose commercial hybrids. [67]. On the

other hand, P. millet icms7709 and Sudan pop I breeds are unable to produce enough yield per unit of limited water, and thus cannot tolerate the arid environments of Jordan.

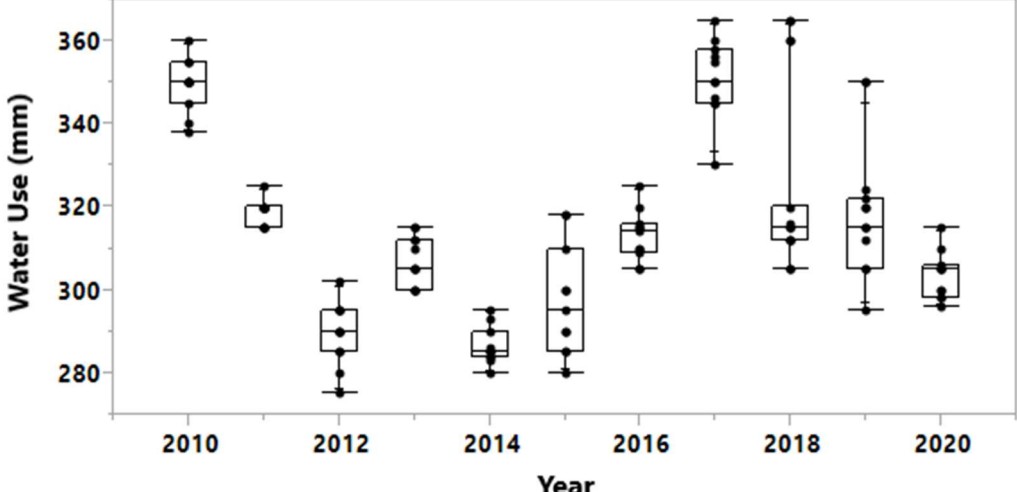

**Figure 8.** Temporal Variability of irrigation water applied regarding the millet breeds.

According to the second scenario, the total fresh and dry biomasses have increased as fodder production has increased due to cutting practices. This suggests that millet breed variability toward cutting is due to the physiological behavior of vegetative growth rather than the surrounding environment. The total yields of millet breeds ranged from 9.9 to 20.2 tons per acre for fresh yield and from 3.3 to 7.7 tons per acre for dry yield. The average WUE of millet breeds in the second scenario ranges from 32.03 to 64.82 kg/ha·mm for fresh biomass and 10.8 to 24.6 kg/ha·mm for dry biomass.

In contrast to the first scenario, millet breeds can show good forage production (i.e., animal feed) as evidenced by insignificant variation in fresh and dry bio-mass and WUE, particularly for the IP19586, IP22269, IP19612, IP7704, and HHVBC tall B6 millet breeds. However, there are differences in the ratio of dry to fresh yield, which ranges from 0.31 at the MC94C2 breed to 0.53 at the HVBC tall B6 millet breed, despite the fact that the season varies greatly (for example, dry conditions). Figures 9 and 10 show that the millet breeds IP19586, IP22269, and IP19612 are significant good producers that can be used for forage production in Jordan's arid environment.

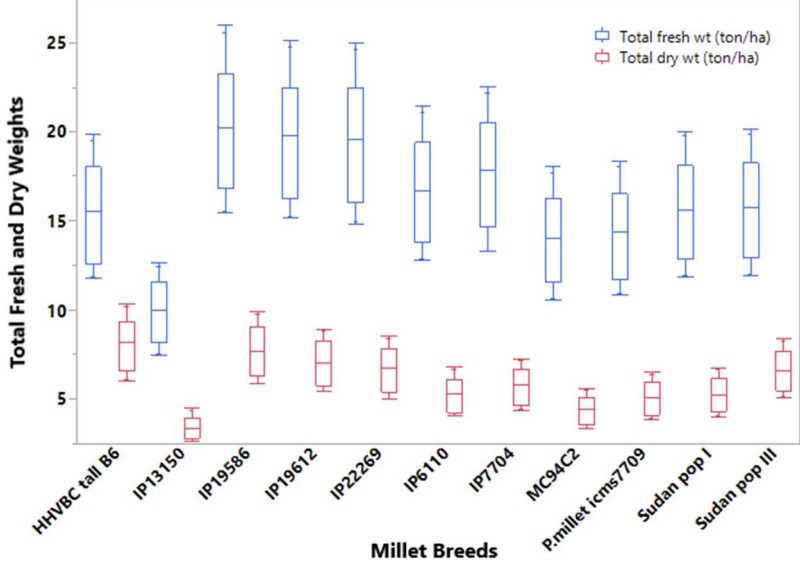

**Figure 9.** Total fresh and dry biomass variabilities regarding millet breeds.

**Table 7.** Comparison between millet production and WUE under different scenarios regarding Millet breeds.

| Millet Breeds | First Scenario | | | | Second Scenario | | | |
|---|---|---|---|---|---|---|---|---|
| | Fresh Yield (ton/ha) | Dry Yield (ton/ha) | Fresh WUE (kg/ha·mm) | Dry WUE (kg/ha·mm) | Total Fresh Biomass (ton/ha) | Total Dry Biomass (ton/ha) | Total WUE Fresh (kg/ha·mm) | Total WUE Dry (kg/ha·mm) |
| IP22269 | 15.46 ab | 4.05 ab | 48.86 a | 12.81 a | 19.39 ab | 6.59 abc | 61.28 abc | 20.83 abc |
| IP19612 | 15.36 ab | 4.23 ab | 49.85 a | 13.58 a | 19.52 ab | 6.94 ab | 62.64 ab | 22.29 ab |
| IP19586 | 15.89 ab | 4.22 ab | 51.06 a | 13.54 a | 20.17 a | 7.66 a | 64.82 a | 24.63 a |
| IP6110 | 9.04 d | 2.27 c | 28.98 c | 7.29 b | 16.64 abc | 5.20 cd | 53.37 abc | 16.68 bcde |
| IP7704 | 10.68 cd | 2.66 c | 34.24 bc | 8.54 b | 17.53 abc | 5.65 bcd | 56.25 abc | 18.14 bcd |
| IP13150 | 17.65 a | 4.60 a | 57.29 a | 14.92 a | 9.86 d | 3.34 e | 32.03 d | 10.83 e |
| HHVBC tall B6 | 8.97 d | 2.33 c | 28.91 c | 7.51 b | 15.37 bc | 8.02 a | 49.56 abc | 25.85 a |
| MC94C2 | 10.29 cd | 2.64 c | 33.39 bc | 8.59 b | 13.91 cd | 4.35 de | 45.16 cd | 14.12 de |
| P.millet icms7709 | 8.62 d | 2.31 c | 27.49 c | 7.38 b | 14.23 cd | 5.07 cd | 45.39 cd | 16.18 cde |
| Sudan pop I | 8.30 d | 2.19 c | 26.96 c | 7.12 b | 15.52 bc | 5.25 cd | 50.41 abc | 17.05 bcd |
| Sudan pop III | 13.80 bc | 3.71 b | 44.58 ab | 12.00 a | 14.63 c | 6.55 abc | 47.21 bcd | 21.14 abc |
| Tukey–Kramer HSD | 3.7276 | 0.8535 | 14.139 | 3.3788 | 4.4631 | 1.5261 | 17.176 | 5.9884 |

Levels not connected by the same letter within the same column are significantly different at 95% confidence level.

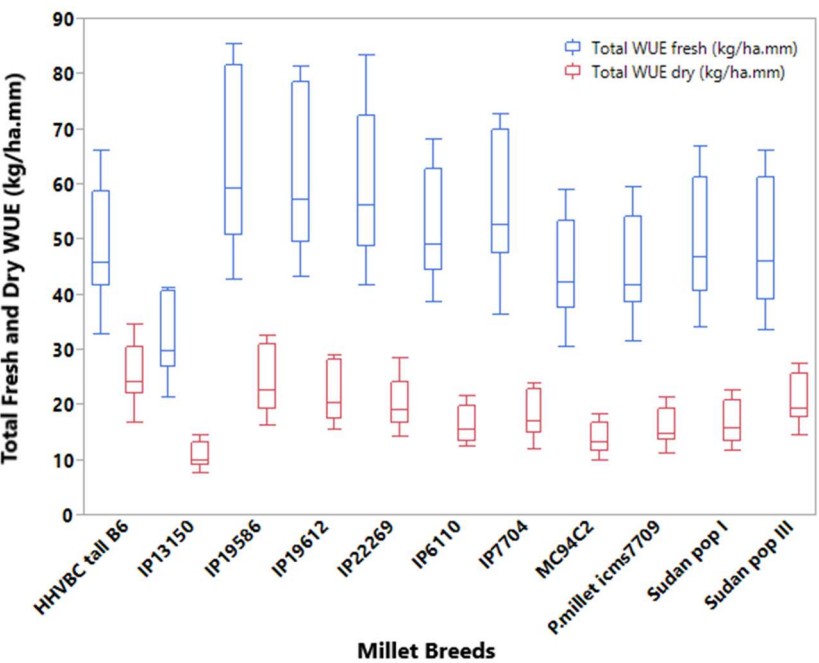

**Figure 10.** WUEs of total fresh and dry biomass regarding millet breeds.

*3.6. Comparison of Seed Parameter among Millet Breeds*

Millet breeds differ in terms of seed production and arid climate tolerance (Table 8). Under the Jordanian climate conditions and regardless of seasonal variations, the HHVBC tall B6 millet breed has the highest seed yield of 0.12 ton/ha and WUE of 0.39 07 kg/ha·mm. As a result, it is able to tolerate drought conditions and reproduce properly. In contrast to the preceding, the IP22269 breed, with the lowest seed yield of 0.02 ton/ha and seed WUE of 0.05 kg/ha·mm, fails to tolerate Jordan's drought conditions and propagate properly.

**Table 8.** Comparison between seed yield, seed WUE and HI means regarding Millet breeds main properties.

| Millet Breed | Seed Yield (ton/ha) | Seed WUE (kg/ha·mm) | HI |
|---|---|---|---|
| IP22269 | 0.0165 f | 0.0522 h | 0.0042 i |
| IP19612 | 0.0665 cd | 0.2130 cd | 0.0157 f |
| IP19586 | 0.0248 f | 0.0798 gh | 0.0059 h |
| IP6110 | 0.0900 b | 0.2887 b | 0.0396 b |
| IP7704 | 0.0451 e | 0.1447 ef | 0.0170 f |
| IP13150 | 0.0258 f | 0.0839 fgh | 0.0056 h |
| HHVBC tall B6 | 0.1212 a | 0.3907 a | 0.0521 a |
| MC94C2 | 0.0847 b | 0.2749 bc | 0.0322 d |
| P.millet icms7709 | 0.0526 de | 0.1679 de | 0.0229 e |
| Sudan pop I | 0.0782 bc | 0.2540 bc | 0.0358 c |
| Sudan pop III | 0.0421 e | 0.1359 efg | 0.0114 g |
| Tukey–Kramer HSD | 0.01626 | 0.06295 | 0.00138 |

Levels not connected by the same letter within the same column are significantly different at 95% confidence level.

In this study, the millet breeds' propagation capability can be categorized into three classes based on HI; (1) the high capability class (HI of above 0.04) as HHVBC tall B6 breed, (2) the moderate capability class (0.02 < HI < 0.04) as IP6110, MC94C2, Sudan pop I, IP22269, IP19612, and P. millet icms7709 breeds, and (3) the poor capability class (HI < 0.02) as IP7704, Sudan pop III, IP13150, and IP19586 breeds. Kumari explained that the high WUE for biomass yield purposes obtained under drought stress is mainly due to the improved HI [68].

### 3.7. Correlation of Drought Indicators with Millet Growth and Production Parameters

The correlation matrix between the millet growth parameters and production with drought indices shows that the *SPI* and *PDI* precipitation drought indices are the most significant indices that can represent the smallest variations between millet breeds, followed by *TDI*. *VDI*, on the other hand, showed little correlation with millet growth parameters and production. This implies that the *CDI* is preferred for combining only the *PDI* and *TDI* indices (Table 9).

**Table 9.** Pearson correlation coefficients between drought indices and crop growth and yield parameters regardless millet breeds.

| | *SPI* | *PDI* | *TDI* | *VDI* | *CDI* |
|---|---|---|---|---|---|
| Water Use (mm) | −0.6278 | −0.6536 | 0.3157 | −0.1792 | −0.5456 |
| Number of Branches | 0.4127 | 0.4965 | −0.2487 | 0.0738 | 0.3737 |
| Plant diameter | 0.4984 | 0.6763 | −0.3496 | 0.0838 | 0.4945 |
| Plant height | 0.4978 | 0.6679 | −0.3635 | 0.1154 | 0.4925 |
| Root Depth | 0.6245 | 0.6728 | −0.3425 | 0.1214 | 0.4852 |
| Fruit Length | 0.4989 | 0.5367 | −0.3236 | 0.0734 | 0.3922 |
| Fruit Diameter | 0.4350 | 0.5515 | −0.3858 | 0.0978 | 0.4167 |
| Frist Scenario Fresh Yield (Ton/ha) | 0.3280 | 0.3754 | −0.2010 | 0.0471 | 0.2712 |
| Frist Scenario Dry Yield (Ton/ha) | 0.3791 | 0.3949 | −0.2119 | 0.0614 | 0.2898 |
| Frist Scenario Fresh Biomass WUE (kg/ha·mm) | 0.3831 | 0.4662 | −0.2221 | 0.0789 | 0.3622 |
| Frist Scenario Dry Biomass WUE (kg/ha·mm) | 0.3260 | 0.4850 | −0.2314 | 0.0893 | 0.3793 |
| Seed Yield (ton/ha) | 0.2951 | 0.2219 | −0.1180 | 0.0397 | 0.1658 |
| Seed WUE (kg/ha·mm) | 0.1365 | 0.2873 | −0.1360 | 0.0582 | 0.2276 |
| HI | −0.0103 | −0.0146 | 0.0080 | −0.0016 | −0.0103 |
| Second Scenario Fresh Yield (ton/ha) | 0.5191 | 0.5239 | −0.3659 | 0.0973 | 0.4012 |
| Second Scenario Dry Yield (ton/ha) | 0.5840 | 0.4475 | −0.3407 | 0.0736 | 0.3301 |
| Second Scenario WUE Fresh Biomass (kg/ha·mm) | 0.6780 | 0.6245 | −0.3932 | 0.1275 | 0.5031 |
| Second Scenario WUE Dry (Biomass kg/ha·mm) | 0.6443 | 0.5477 | −0.3609 | 0.1028 | 0.4296 |

According to the correlation matrix, there is a strong relationship between the amount of water used and the precipitation and drought indices. The drier the environmental conditions, the more water required to compensate for the soil–water–plant deficit. However, most of the applied water under drought conditions does not necessary uptake by the plant rather than being evaporated from the soil system.

As a result, the water use term is not recommended for correlating plant growth and yield factors because this flawed expression may lead to the crop being less water efficient when it could be the most promising climate-resilient crop. As a result, the water use term is not recommended for correlating plant growth and yield factors because this flaw expression may lead to the crop being less water efficient when it could be the most promising climate resilient crop. Furthermore, the *SPI* and *PDI* indices show a moderate ($0.3 \geq r \geq 0.6$) correlation with all other millet growth parameters and yield, indicating that millet is susceptible to drought conditions. According to the matrix, pearl millet tolerates drought conditions significantly by increasing root elongation and density to increase soil–water uptake from deeper soil layers. Furthermore, its phenological characteristics improve its drought tolerance by reducing leaf area and adjusting osmotic potential. A similar explanation was found by Cattivelli et al. pearl millets holding morpho-physiological traits can tolerate drought through stomatal conductance, photosynthetic capacity, the

timing of the phenological phases, stem reserve mobilization in drought stress, reduced leaf area, rooting depth and density, cuticular resistance and surface roughness, osmotic adjustment, membrane composition, antioxidative defense, and accumulation of stress-related proteins [69].

The *TDI* indicator was able to moderately correlate with water use, plant diameter and height, root depth, and fruit length and diameter, which could be attributed to crop physiology influences during heatwaves or extreme temperatures. The presence of higher-than-average temperatures simulates stress factors in millet, increasing its ability to withstand stress. According to others, Pearl millet is a hardy, climate-smart grain crop that is ideal for environments prone to drought and heat stress [43,48]. According to Dai, the millet phenological character compensates for its potential in case of any stress [4]. Depending on the critical stage occurrence during the plant growth stages, the millet breeds impact differently. According to Aparna et al., daytime maximum temperatures above 42 °C, as well as the associated vapor pressure deficit (VPD) during flowering time, directly reduce seed setting in pearl millet. [62]. Increased air temperature, on the other hand, may result in decreased reproduction due to heat damage to the pollen grain and increased sterility [70–72].

The precipitation and heat-stress drought indices have a moderate correlation with the WUEs, indicating that millets can withstand climatic changes to varying degrees depending on millet variety. The WUE is more influenced by edaphic than climatic conditions [64]. Furthermore, there are several categories of millets: (1) Agronomy-related traits, that is known as yield and yield components; (2) Morphology-related traits that are drought tolerance through changing its morphological and biochemical properties as flag leaf and leaf tensile strength; (3) Physiology-related traits that tolerate drought through osmotic adjustment; and (4) Biochemical-related traits that produce antioxidants to cope with unfavorable reactive oxygen species during stress conditions [49]. Finally, according to Sultan et al., photoperiod-sensitive pearl millet traditional cultivars appear to be more resistant to future climate conditions than improved cultivars with high genetic yield potential [73]. As a result, it is recommended that pearl millet cultivation in Jordan be strategically managed and adapted by recognizing production options in light of anticipated climate change scenarios.

## 4. Conclusions

Some pearl millet breeds demonstrated moderate to high resilience to extreme environmental conditions, particularly insufficient moisture and heat stress, and thus can withstand climate change. Although there are differences in millets breeds' responses to drought and heat stresses, as evidenced by differences in growth and plant production indicators, they have proven to adapt to the arid Jordanian environment and thus sustain their productivity in drought-prone conditions.

The IP19586, IP22269, and IP19612 millet breeds are preferred for vegetative production because they provide good forage support. Their total yields were approximately 19 ton/ha fresh yield and approximately 7 ton/ha dry yield, which is twice as much as other millet breeds. Furthermore, the breeds mentioned above were bred to withstand drought conditions in Jordan, where they provided the highest WUEs (above 60 kg/ha·mm) for fresh biomass and (above 20 kg/ha·mm) for dry biomass. In terms of propagation characteristics, the HHVBC tall B6 breed exhibits good seed propagation with an average production of 0.12 ton/ha and a WUE of 0.39 07 kg/ha·mm, whereas the IP22269 breed demonstrated significantly lower seed yield, indicating a failure to propagate properly in arid-drought conditions. The correlation of millet growth and drought indices (*VDI*) revealed a very weak relationship (r < 0.3) and thus cannot be used for drought monitoring in Jordan. Temperature and precipitation drought indices both had strong correlations with millet growth factors (r > 0.6) and a moderate correlation with yield factor (0.3 < *r* < 0.6). As a result, the findings of this study suggest that a combined precipitation–temperature drought index be used to investigate, monitor, and assess millet growth parameters and production under climatic stress. Furthermore, based on these findings, the authors rec-

ommend that other drought indices be investigated in order to develop a pearl millet monitoring plan.

Despite the fact that this study demonstrated that pearl millet breeds were able to tolerate drought conditions significantly due to their phenological characteristics, and thus set as a climate smart grain crop in Jordan, more research and replications are needed to investigate spatial and temporal variability under various Jordan climate zones.

**Author Contributions:** Conceptualization, N.B.H., F.J.A. and M.I.A.-Q.; Methodology, N.B.H.; Investigation, N.B.H.; Writing—Original Draft, F.J.A.; Writing—Review & Editing, F.J.A. and M.I.A.-Q. Resources, F.J.A. and M.I.A.-Q.; Supervision, F.J.A. and M.I.A.-Q. All authors have read and agreed to the published version of the manuscript.

**Funding:** This research received no external funding.

**Institutional Review Board Statement:** Not applicable.

**Informed Consent Statement:** Not applicable.

**Data Availability Statement:** Data sharing not applicable to this article as no datasets were generated or analyzed during the current study.

**Acknowledgments:** The authors would like to acknowledge the logistic support provided by the National Agriculture Research Center for conducting this research.

**Conflicts of Interest:** We have no pecuniary or other personal interest, direct or indirect, in any matter that raises or may raise a conflict with our duties as authors of the article entitled (Investigating the Pearl Millet (*Pennisetum glaucum*) as a climate-smart crop under drought tolerance under Jordanian arid environments. Authors: Nabeel Bani Hani, Fakher J. Aukour, Mohammed I. Al-Qinna).

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
