# Peer review of "Investigating the Pearl Millet (Pennisetum glaucum) as a Climate-Smart Drought-Tolerant Crop under Jordanian Arid Environments"

_sustainability, doi:10.3390/su141912249_

Round 1

Reviewer 1 Report

In general, the article is well written and nicely designed.

1. The abstract must be clear and reflect core findings along with brief methodology and recommendations, kindly reframe it.

2. Introduction and methodology are ok, however, the hypothesis and objective of the study must be clear, kindly revise.

3. Results and discussion are ok but results must be supported by some latest findings.

4. Conclusion should be clear and crisp.

5. Language check is suggested. Check the references critically.

Author Response

Reviewer # 1

General Point: the article is well written and nicely designed.

Response: Thanks for your nice words

Comment 1: The abstract must be clear and reflect core findings along with brief methodology and recommendations, kindly reframe it.

Response: Reviewers totally agree. The abstract is revised to reflect the core findings and recommendations along with the adopted methodology. Please see the new abstract.

For climate change adaptations with water scarcity, Pearl Millet was used in arid climatic conditions, a field randomized complete block design experiment of 11 species of (HHVBC tall B6, IP13150, IP19586, IP19612, IP22269, IP6110, IP7704, MC94C2, P. millet icms7709, Sudan pop I, and Sudan pop III) were tested at Deir-Alla Regional Agriculture Research center at the middle Ghor within Jordan Valley. Plants deficit irrigation of 80% was maintained based on crop water requirements using a time-domain reflectometer. Plant morphological characteristics, forage production, seed formation, and WUE were monitored for two case seed and forage production scenarios for ten years. Water use was measured using a water balance equation modified with soil deficit monitoring at various soil depths. WUE was calculated for three replicates per year. Individual and combined drought indices of precipitation, temperature, and vegetation were calculated and correlated with millet morphological and yield parameters. Climate change analyses show significant impacts reaching a 1 mm/y reduction in precipitation and a 0.04 mm/y increase in air temperature, subjecting the study area to become more prone to drought events. Along with proven drought intensity increase over time as indicated by drought indices temporal analyses, Millet species showed significant drought tolerance capacities and capabilities to grow under arid drought-prone conditions through adjusting their system to tolerate salt, heat, and water stresses. Under the seed production scenario, the WUE ranged from 27 to 57.3 kg/ha.mm and from 7.1 to 14.9 kg/ha.mm for fresh and dry conditions, respectively. The IP13150 millet species showed the highest values and thus could potentially be a good substitute in water scarcity situations with an average production of 17.7 ton/ha, while P. millet icms7709 and Sudan pop I. millet species were unable to tolerate drought environments of Jordan. For the vegetative production scenario, the WUE ranged from 32.03 to 64.82 kg/ha.mm for the fresh biomass, and from 10.8 to 24.6 kg/ha.mm for the dry biomass, respectively. The highest WUEs were associated with IP19586, IP22269, IP19612, IP7704, and HHVBC tall B6 millet species due to their phenological characteristics to tolerate drought and heat conditions. Contrary to the vegetation drought index, both precipitation and temperature drought indices showed strong correlations (above r>0.6) with plant growth factors and moderate correlation (0.3<r<0.6) with yield factors, suggesting the potential of combined precipitation and temperature to be used for monitoring and selection of drought-tolerance crops.

Comment 2: Introduction and methodology are ok, however, the hypothesis and objective of the study must be clear, kindly revise.

Response: Totally agree. The following hypothesis was added:

“The main hypothesis is that Drought Indicators can be used as a selection criterion to identify and monitor Pearl Millet drought tolerance capacity”.

Also, the objective is revised to clearly define the main goal of this research, as follows:

“The main objective of this study was to correlate the growth and yield factors of different pearl millet varieties with several drought indicators.”

Comment 3: Results and discussion are ok but results must be supported by some latest findings.

Response: Partially agree. The results were revised and supported by the latest findings especially as related to crop yield production and morphological responses. However, correlating the drought indices with millet growth and production is unique and lacks of similar conclusions.

Comment 4: The conclusion should be clear and crisp.

Response: The conclusions were revised to be more specific, clear, and crisp as much as possible. Please see the new revised version.

Comment 5: Language check is suggested. Check the references critically.

Response: The manuscript language was checked and revised. Also, the references were checked. Please see the new revised version.

Reviewer 2 Report

In a changing climate, the study conducted by the authors is relevant and timely.

Literature review sufficiently reveals the essence of the problem. The authors cited 74 literary sources, mostly publications not older than 10 years.

The material is presented in a well-structured form.

The research methods are described in detail. The results are scientifically sound and obtained according to the methods used. The results are convincingly illustrated with 9 tables and 10 figures. Statistical processing of the data was carried out.

Morphological and physiological parameters of millet resistance to drought were comprehensively studied.

The data are discussed, references to scientific works of other authors are used.

The scientific hypothesis is confirmed by the results.

The authors recommend using precipitation drought index and temperature drought index or using a combined drought index.

The conclusions are consistent with the experimental data obtained.

Author Response

Reviewer # 2

Comment 1: In a changing climate, the study conducted by the authors is relevant and timely.

Response: Climate change has been tackled carefully to ensure its relevance to the pearl millet drought adaptation plan.

Comment 2: The literature review sufficiently reveals the essence of the problem. The authors cited 74 literary sources, mostly publications not older than 10 years.

Response: The literature language was revised. The objective was revised and included a hypothesis.

Comment 3: The material is presented in a well-structured form.

Response: Thanks for your nice words

Comment 4: The research methods are described in detail.

Response: The methodology was revised to include the plant sampling and testing procedures.

Comment 5: The results are scientifically sound and obtained according to the methods used. The results are convincingly illustrated with 9 tables and 10 figures. Statistical processing of the data was carried out.

Response: The discussion was enriched with more comparisons.

Comment 6: Morphological and physiological parameters of millet resistance to drought were comprehensively studied.

Response: Thanks, this is true.

Comment 7: The data are discussed, and references to scientific works of other authors are used.

Response: As mentioned, the discussion was more elaborated than before at the revised version.

Comment 8: The scientific hypothesis is confirmed by the results.

Response: Yes, totally agree. And also, a hypothesis was inserted for a better definition of the paperwork and scope.

Comment 9: The authors recommend using precipitation drought index and temperature drought index or using a combined drought index.

Response: That is true. The vegetation index was weak. The authors recommended a top investigation for further drought indices.

Comment 10: The conclusions are consistent with the experimental data obtained.

Response: The conclusions were revised to become crisper.

Reviewer 3 Report

Please take your time and review your manuscript again.  There are a significant number of typos and editing needs for the paper.  By so doing, your manuscript will improve in terms of the presentation and quality.

Below are some common mistakes observed through the manuscript.

1.       Before you use an abbreviation, please define it earlier e.g. GOJ (Line 63)

2.       Write the latitudes and longitudes well “latitude of 32o13’E” and longitude of 35o37’N (Line 124

3.       The first letter of the first word of a new sentence should start with a capital letter (e.g. Lines 348 and 427)

4.       Avoid starting a new sentence with the reference number.  Instead say Bidinger et al (12) (e.g. Line 403)

5.       There was no description of how the plant measurements were done.

Author Response

Reviewer # 3

General Comment: Please take your time and review your manuscript again.  There are a significant number of typos and editing needs for the paper.  By so doing, your manuscript will improve in terms of presentation and quality. Below are some common mistakes observed in the manuscript.

Comment 1: Before you use an abbreviation, please define it earlier e.g. GOJ (Line 63)

Response: All abbreviations were checked and defined correctly.

Comment 2: Write the latitudes and longitudes well “latitude of 32o13’E” and longitude of 35o37’N (Line 124).

Response: The Centre is located at a latitude of 32°13’E and longitude of 35°37’N and 224 below sea level (Figure 1).

Comment 3: The first letter of the first word of a new sentence should start with a capital letter (e.g. Lines 348 and 427).

Response: Totally agree. All sentences were revised carefully.

Comment 4: Avoid starting a new sentence with the reference number.  Instead, say Bidinger et al (12) (e.g. Line 403).

Response: Totally agree. The manuscript was revised to ensure such mistakes are avoided.

Comment 5: There was no description of how the plant measurements were done.

Response: Totally agree. A sentence was added describing the plant sampling. A destructive 1.5m×1.5m Quadrat sampling method was used to collect plant samples after maturing. Plant aboveground samples were randomly selected from each plot, chopped, fresh weighed, oven dried at 80°C for 24 hours, and finally dry weighed. In addition, millet seeds were collected by hands, fresh weighed, dried using a continuous flow drier for 48 hours, and finally dry weighed.

Reviewer 4 Report

This study investigated the responses of different pearl millet cultivars to different climatic conditions. Although the research topic is more meaningful, there are major problems in the writing and organization of the thesis. Especially the language problem, which uses a lot of passive voice (was, were), makes the manuscript less readable. In addition, the figures are not standardized and have not reached the level of publication.

1. In the abstract section, the main findings of this study are not highlighted.

2. References are not listed in order.

3. Line 288. The meaning of each parameter in the equation needs to be given.

4. What do the horizontal lines in Fig. 4 and Fig. 5 represent.

5. Table 5, In the display of results, it is not necessary to display all the results, but to summarize.

Author Response

Reviewer 4

Comment 1: This study investigated the responses of different pearl millet cultivars to different climatic conditions. Although the research topic is more meaningful, there are major problems in the writing and organization of the thesis. Especially the language problem, which uses a lot of passive voice (was, were), makes the manuscript less readable.

Response: Authors totally agree. The manuscript was revised by an English Native Editor.

Comment 2: In addition, the figures are not standardized and have not reached the level of publication.

Response: Authors revised the figures to become consistent, standardized, and reach the level of publication

Comment 3: In the abstract section, the main findings of this study are not highlighted.

Response: The authors revised the abstract and highlighted the study's main findings. Please check the new abstract.

Comment 4: References are not listed in order.

Response: Authors apologize for this mistake. References were revised and listed in order.

Comment 5: Line 288. The meaning of each parameter in the equation needs to be given.

Response: Authors appropriately revised all equation parameters (especially Eq. 12). 

Comment 6: What do the horizontal lines in Fig. 4 and Fig. 5 represent?

Response: The horizontal lines in Figures 4 and 5 represent the Drought severity classes. The authors added the classes to each figure for better illustration.

Comment 7: Table 5, In the display of results, it is not necessary to display all the results, but to summarize.

Response: Authors agree. The table was revised to represent the experiment period with an indication of drought severity and class.

Reviewer 5 Report

The paper assesses the possibilities of adaptation to drought of pearl millet varieties. The impact of drought was assessed on
morphological and yield-forming reactions of eleven cultivars of this plant.

The article should be checked by a native speaker, because it has a lot of linguistic errors.

L90: Latin plant names should be written in italics.

L183: Why are there asterisks * next to the abbreviations?

Eleven varieties were tested in this study, Authors sometimes use species, accessions or varieties of pearl millet.
Please be consistent and use the right name.

Table 4. Please explain the abbreviations used in the table. In my opinion, tables and figures should be self-explanatory.

The quality of figures should also be improved. They are unfortunately blurred.

The work is sometimes unclear and written carelessly.

It should be explained in which years the millet experiment was carried out.
Was it in the same period as the weather data was collected?
There are different time periods in different figures. In my opinion, the work should be better described.
The experimental material is valuable and, when better processed, can make a valuable contribution to the knowledge of drought.

Author Response

Reviewer 5

Comment 1: The paper assesses the possibilities of adaptation to the drought of pearl millet varieties. The impact of drought was assessed on the morphological and yield-forming reactions of eleven cultivars of this plant. The article should be checked by a native speaker because it has a lot of linguistic errors.

Response: Authors totally agree. The manuscript was revised by an English Native Editor.

Comment 2: L90: Latin plant names should be written in italics.

Response: Authors totally agree. The name is written in italic.

Comment 3: L183: Why are there asterisks * next to the abbreviations?

The (*) marks represent the modified parameters. Those are explained by equations 5, 6, 7, and 8.

Comment 4: Eleven varieties were tested in this study, Authors sometimes use species, accessions, or varieties of pearl millet. Please be consistent and use the right name.

Response: Authors apologize for these mistakes. Actually, they represent breeds. The authors revised the manuscript and corrected all names.

Comment 5: Table 4. Please explain the abbreviations used in the table. In my opinion, tables and figures should be self-explanatory.

Response: Authors revised the table. Abbreviations are removed to become self-explanatory.

Comment 6: The quality of the figures should also be improved. They are unfortunately blurred.

Response: Authors revised all figures to become consistent, standardized, and reach the level of publication.

Comment 7: The work is sometimes unclear and written carelessly.

Response: Authors revised the manuscript statement to provide meaningful clear statements for the readers.

Comment 8: It should be explained in which years the millet experiment was carried out. Was it in the same period as the weather data was collected? There are different time periods in different figures. In my opinion, the work should be better described.

Response: Authors understand the conflict between different timing and representations. In general, climate assessments require 30 years or more of historical data to have a real indication of drought severity based on a long-time average. In our case study, the climate period used was 30 years starting from 1980 to 2020. On the other hand, the experiment was conducted for 10 consecutive years starting from 2010 to 2020. The correlations between Millet responses and drought indices are only for the ten experiment years. The authors adjusted all figures and tables to become more consistent and justified the variations in the methodology section.

Comment 9: The experimental material is valuable and, when better processed, can make a valuable contribution to the knowledge of drought.

Response: Authors appreciate your point of view. The manuscript undertook several revisions. The authors hope it reached the desired standards.

Round 2

Reviewer 4 Report

On the whole, the author has made serious revisions according to the review comments, and the quality of the figures is still a little lacking, and the journal editor can give specific comments.

Reviewer 5 Report

The authors made corrections to the article.